# Burning Mouth Syndrome and Hypertension: Prevalence, Gender Differences and Association with Pain and Psycho-Social Characteristics—A Case Control Study

**DOI:** 10.3390/ijerph20032040

**Published:** 2023-01-22

**Authors:** Daniela Adamo, Federica Canfora, Elena Calabria, Noemi Coppola, Mattia Sansone, Gianrico Spagnuolo, Giuseppe Pecoraro, Massimo Aria, Luca D’Aniello, Michele Davide Mignogna, Stefania Leuci

**Affiliations:** 1Department of Neuroscience, Reproductive Sciences and Odontostomatology, University of Naples Federico II, 80138 Naples, Italy; 2Department of Health Sciences, School of Dentistry, University Magna Graecia of Catanzaro, 88100 Catanzaro, Italy; 3Therapeutic Dentistry Department, Institute for Dentistry, Sechenov University, Moscow 119991, Russia; 4Department of Economics and Statistics, University Federico II of Naples, 80138 Naples, Italy; 5Department of Social Sciences, University Federico II of Naples, 80138 Naples, Italy

**Keywords:** burning mouth syndrome, hypertension, pain, mood disorder, systemic comorbidities

## Abstract

Background: To assess the prevalence of hypertension (HTN) in burning mouth syndrome (BMS) patients and to investigate its relationship with sociodemographic factors, pain and the psychological profile. Methods: A case-control study was conducted by enrolling 242 BMS patients and 242 controls matched for age and gender. Sociodemographic and clinical characteristics were recorded, and all participants completed numeric rating scale (NRS), the short-form of the McGill pain questionnaire (SF-MPQ), the Hamilton rating scale for anxiety and depression (HAM-A, HAM-D), the Pittsburgh sleep quality index (PSQI) and the Epworth sleepiness scale (ESS). Results: The BMS patients presented with a statistically significant higher prevalence of HTN compared to that in the controls (55% versus 33.5%; *p*-value: <0.001) and higher median scores of the NRS, SF-MPQ, HAM-A, HAM-D, PSQI and ESS (*p* < 0.001). Multivariate regression analysis in the BMS patients indicated positive correlations between HTN and age, systemic diseases, drug consumption and anxiety (*p*-value: <0.001) and these predictors were responsible for 11.3% of the HTN variance in the BMS patients, when considered together. Conclusions: The prevalence of HTN was significantly higher in the BMS patients, since ageing, the presence of comorbidities, drug consumption and anxiety were potential predictors. Further studies are needed to better investigate the relationship between BMS and HTN.

## 1. Introduction

Hypertension (HTN) is defined as a repeatedly elevated blood pressure (BP) exceeding 140/90 mm Hg and represents the leading cause of cardiovascular disease and premature death [1]. It affects approximately one billion people and increases with age, with the prevalence in individuals between 20 and 44 being about 26%, reaching up to 78% in the geriatric population (over 65) [1,2]. Midlife HTN (45–65) is considered an important contributor to late-life dementia and Alzheimer’s disease (at an age greater than 65) [3,4]. The individual’s genetic profile, including insufficient physical activity, obesity, dyslipidemia, diabetes and a high-fat diet, increases the risk of developing HTN [5,6].

Burning mouth syndrome (BMS) is chronic and debilitating oral pain of the normal oral mucosa. The overall prevalence of BMS in clinical practice is on average 4%, reaching a prevalence of 18% in post-menopausal women [7]. Its etiopathogenesis remains unclear and is probably of a multifactorial origin, with growing evidence that BMS may be a neuropathic disorder, with an increase in central pain sensitization processes and a reduction in the functioning of the descending pain inhibitory mechanisms [7,8]. BMS is classified as an idiopathic orofacial pain with or without somatosensory changes in the International Classification of Orofacial Pain (ICOP 2020) [9]. The diagnosis of BMS has evolved from basic intraoral exclusion screening to extensive clinical and laboratory investigations [10,11], which include the screening of comorbidities and other chronic pain and somatosensory testing. The most common site for BMS is the tongue (anterior two-thirds or tip), followed by the hard palate, gingivae, lower lips and pharynx. It usually occurs bilaterally and symmetrically [12,13,14]. Additional subjective oral symptoms, such as dysgeusia, itching, xerostomia and sialorrhea have been reported [15,16].

It is known that there are functional interactions between pain and the cardiovascular system, which may be different between healthy subjects and patients with chronic pain [17]. Indeed, in healthy subjects, through the activation of the sympathetic nervous system, the acute painful stimuli can induce a blood pressure increase, which in turn may stimulate the baroreceptors, activating the descending pain inhibitory pathways [18,19]. This mechanism suggests an inverse relationship between higher blood pressure and acute pain perception in healthy subjects [20].

Instead, some studies have found that in patients suffering from chronic pain conditions, such as temporomandibular disorders, low back pain and fibromyalgia, the elevated resting blood pressure level is associated with an increased sensitivity to pain, suggesting a dysfunction of the blood pressure/pain regulatory feedback loop [21,22,23]. These dysfunctions may be due to a reduction in baroreceptor sensitivity, which is not able to activate the pain inhibitory pathways or leads to direct impairment of the pain inhibitory pathways, not activated by a normal increase in baroreceptor stimulation [22].

Regarding BMS, the relationships between elevated blood pressure and pain and the prevalence of HTN in this disease are not completely clear [24,25]. In the study of De Pedro et al., based on a sample set of 20 BMS patients and 40 controls, the authors reported that BMS patients presented with a worsened health status, suffering from a higher number of comorbidities and consumed more medications compared with controls [8]. Instead, in the case control study of JQ Jin et al. on 352 BMS patients and 391 controls, despite significant differences between groups being found in terms of gender, age, BMI, education level, job status, brain imaging abnormalities, total cholesterol, scores of anxiety and depression, no differences were found in the prevalence of HTN between cases and controls [26].

Recently, in a sample of 100 BMS patients, HTN was found in 58% of the subjects and it was considered the most important predictor of white matter changes (WMCs) of the brain, which in turn may potentially amplify pain perception in patients affected by this disease [27].

Therefore, we wanted to further evaluate the prevalence of HTN in a large sample of BMS patients, compared with that in a control group of subjects without BMS, in order to compare our results with previous studies and to better elucidate the relationships between HTN and BMS, analyzing if this comorbidity may be an aggravating factor that should be investigated for BMS assessments.

To the best of our knowledge, none of these previous studies have been focused specifically on the analysis of the prevalence of HTN in a sample of BMS patients.

In this study, despite BMS predominantly affecting women with a female/male ratio of 3:1 [7], the same number of male and female subjects were enrolled in order to eliminate sex-related differences in the prevalence of HTN in BMS patients, which remains unclear; indeed, it is well known that the prevalence of HTN is higher in men than in women prior to the onset of menopause, while this increases in women after menopause with a sex-convergence of HTN prevalence later in life [2,28]. 

The endpoints of the current study were:
-to investigate the prevalence of HTN in a large sample of BMS patients, compared with that in a control group of control subjects matched for age and gender;-to analyze the differences in HTN prevalence between males and females, exploring this prevalence at different ages;-to identify the potential predictors of HTN in BMS patients, taking into account the sociodemographic profile (age, employment and marital status), the body mass index (BMI), risk factors (smoking and alcohol use), other systemic comorbidities, drug consumption, pain evaluation and psychological factors.

## 2. Materials and Methods

### 2.1. Study Design

A case-control study was carried out over a period from April 2020 to January 2022 at the Oral Medicine Department of the University of Naples “Federico II”, after obtaining the approval from the Ethical Committee of the University (Approval Number: 251/19—the date of approval being 20 February 2019). The study was conducted in accordance with the ethical principles of the World Medical Association Declaration of Helsinki, and the adopted methods conformed with the Strengthening of the Reporting of Observational Studies in Epidemiology (STROBE) guidelines for observational studies [29].

### 2.2. Data Collection

All potentially eligible participants were invited to participate in the study, and written informed consent was obtained from all subjects involved in the study. First, 250 patients with BMS of either gender and aged >18 years were screened and enrolled in the study. However, only 242 individuals, 121 males and 121 females, met the inclusion and exclusion criteria (Figure 1: flow chart). Based on the average age and gender distribution of the BMS patients, control subjects were then enrolled using convenient sampling in order to obtain controls matched by gender and age. No payment was provided for participation.

### 2.3. Inclusion and Exclusion Criteria

The inclusion criteria for the BMS patients, in accordance with the International Classification of Orofacial Pain (ICOP 2020) 1st edition [9], were: patients experiencing an intraoral burning or dysesthetic sensation, recurring daily for more than two hours per day for more than three months, without any evident causative lesions on clinical examination and investigation; patients with normal blood test findings (including blood count, blood glucose levels, glycated hemoglobin, serum iron, ferritin and transferrin). In addition, patients who had been on the medications for at least 6 months before the onset of oral burning symptoms were admitted, while patients whose chronic therapies had been modified within the 6 months prior to the onset of BMS were not recruited. The dechallenge and rechallenge test was used in all subjects with a suspicion of an adverse drug reaction. 

Control subjects presenting at the dental clinic of the same university for routine dental care were consecutively enrolled according to the following criteria: patients with a normal oral mucosa; patients with normal blood test findings; and patients who had never experienced any burning or dysesthetic sensation of the oral mucosa. 

From both groups, participants were excluded in the case of [30,31,32]: ongoing treatment with psychotropic drugs; drug addiction; a history or presence of oncological or autoimmune disorders; the presence of obstructive sleep apnea syndrome (OSAS); or an incapacity to understand or complete the questionnaires.

### 2.4. Procedures

Each subject underwent intraoral and extra-oral screening by an expert clinician in oral medicine and was progressively enlisted for a routine clinical evaluation, including a medical analysis. The patients were assessed regarding the oral symptoms, and the sites involved, age, years of education, family situation, job status, risk factors (smoking status and alcohol consumption), medical comorbidities and systemic drugs taken. Blood pressure (BP) was obtained in a standardized fashion according to the scientific statement from the American Heart Association [33]. The BP was calculated as the mean of two measurements recorded by a physician. We defined HTN as having a systolic blood pressure 140 mm Hg or greater, a diastolic blood pressure 90 mm Hg or greater or taking medication for HTN [1]. Moreover, we used the measured weight and height in order to calculate the BMI as weight (kilograms) divided by the square of height (meters). In accordance with the WHO classification, the cut-off points considered were: 18.5–24.9 kg/m^2^ for a normal weight, 25.0–29.9 kg/m^2^ for overweight, and >30 kg/m^2^, for obesity, in particular, obesity class I—BMI 30 to 34.9 kg/m^2^, obesity class II—BMI 35 to 39.9 kg/m^2^, obesity class III—a BMI greater than or equal to 40 kg/m^2^ (also referred to as severe, extreme or massive obesity) [34,35].

### 2.5. Pain, Psychological Assessment and Sleep Assessment

The evaluation of pain included the administration of the Numeric Rating Scale (NRS) [36] and the short-form Mc-Gill pain questionnaire (SF-MPQ) [37]. The NRS is a unidimensional scale used to evaluate the pain intensity through a range of scores from 0 to 10 [38]. Instead, the SF-MPQ measures the qualitative aspect of pain with a sensory subscale (11 words) and an affective subscale (4 words) in which the patient has to specify, with a score from 0 to 3, how suitably each word reflects the pain characteristic [39].

The psychological evaluation included the administration of the Hamilton depression rating scale (HAM-D) [40] and the Hamilton anxiety rating scale (HAM-A) [41]. The HAM-D is a scale used to measure the presence and severity of depressive symptoms in clinical practice and research and is administered by a trained physician (GP). The cut-off scores considered are: a score between 7 and 17 indicates mild depression, a score between 18 and 24 indicates moderate depression and a score greater than 24 indicates severe depression. The HAM-A is a scale containing 14 items useful to evaluate both somatic anxiety and psychic anxiety. A total score less than 17 indicates mild anxiety, between 18 and 24 indicates mild-to-moderate anxiety, and between 25 and 30 indicates moderate-to-severe anxiety [42].

The sleep evaluation included administration of the Pittsburgh Sleep Quality Index (PSQI) [43] and the Epworth Sleepiness Scale (ESS) [44]. The PSQI assesses sleep quality and disturbances over a one-month time interval, evaluating seven components, each scored from 0 to 3: subjective sleep quality, sleep latency, sleep duration, habitual sleep efficiency, sleep disturbances, use of sleeping medication and daytime dysfunction. A global PSQI score greater than 5 indicates poor sleep quality [45]. The ESS evaluates the sleep propensity in daily life through eight items, each scored from 0 to 3. In this scale, a higher score corresponds to greater daytime sleepiness [44].

### 2.6. Statistical Analysis

The sample size, equal to 242 patients for each group (the BMS patients and controls), was calculated to obtain a power test value (1-Beta) at no less than 99%, associated with a significance of no more than 1%. This sample size was obtained using the effect size value equal to 0.65, measured in a previously published study regarding age-related white matter changes and the correlation with BMS [27]. The calculations were computed using GPower software (v3.1.9) Faul, F., Erdfelder, E., Buchner, A. et al. Statistical power analyses using G*Power 3.1: Tests for correlation and regression analyses. Behavior Research Methods 41, 1149–1160 (2009). https://doi.org/10.3758/BRM.41.4.1149 [46]. Data analyses were performed using R software (v. 4.2.0—R Core Team, 2016). Faul, F., Erdfelder, E., Buchner, A. et al. Statistical power analyses using G*Power 3.1: Tests for correlation and regression analyses. Behavior Research Methods 41, 1149–1160 (2009). https://doi.org/10.3758/BRM.41.4.1149 Descriptive statistics, including means, standard deviations (SDs), medians and interquartile ranges (IQRs), were calculated to summarize the socio-demographic and clinical characteristics. Fisher’s exact test was used to assess any significant differences between the frequencies of systematic diseases, drug consumption, antihypertensive drugs, oral symptoms, sites involved and clinical parameters (psychological profile, sleep assessment and pain assessment) between the BMS patients and healthy subjects. Instead, the analysis of multiple comparisons between the median values was computed with the Mann–Whitney U test. A dependence analysis between the BMS patients with HTN and the controls and between HTN and the qualitative predictors was performed.

A series of multivariate logistic regression models was computed in order to estimate the presence of potential predictors of HTN in the BMS group and in the healthy subjects. The odds ratio (Bland & Altman, 2000) [47] of each regression model was calculated by considering the socio-demographic variables, smoking, alcohol use, BMI pain intensity and quality (NRS and SF-MPQ), the psychological profile (HAM-A and HAM-D) and the quality of sleep (PSQI and ESS). Next, a full model analysis was computed with all the variables included simultaneously in the model to determine their relative contributions. In detail, a sequential logistic regression model analysis including the predictors, inserted one by one, was performed to obtain unadjusted coefficient estimations. Finally, a full model analysis considering all the predictors simultaneously was carried out to estimate the adjusted coefficients.

In all analyses, the level of significance after the Bonferroni correction was set for multiple tests.

## 3. Results

In total, 242 participants (121 male and 121 female), matched for sex and age, were included in each group. Table 1 summarizes the characteristics of the participants. In particular, there were statistically significant differences considering the education level, employment status and risk factors. Indeed, the mean years of education, as well as the percentage of employment, was higher in the control group (*p*-value: <0.001 **, 0.002 **). In addition, the majority of the BMS patients were non-smokers (181; 74.8%) but included a statistically significant higher proportion of heavy smokers (>15 cigarettes, 28 subjects: 11.6%; *p*-value: 0.007 **). Moreover, there was a statistically significant higher percentage of non-habitual alcohol consumers among the BMS patients and fewer consumers of 1–2 units of alcohol (*p*-value: <0.001 **) compared to numbers in the control group. The frequency distributions of the participants in relation to the BMI categories revealed that overall, the BMS group showed a considerably higher BMI than the healthy controls (*p*-value: 0.001 **) especially with regard to the overweight and class I-II obesity categories. With respect to the female participants, 106 BMS patients and 102 healthy women were in the menopause with no statistical differences between the two groups (*p*-value: 0.783). The comparisons between the median scores of the clinical parameters showed that overall, the BMS patients presented with statistically significantly higher median scores for pain (NRS, SF-MPQ), anxiety (HAM-A), depression (HAM-D) and sleep (PSQI, ESS) compared to those in the healthy subjects (*p*-values: <0.001 **).

Table 2 shows the comorbidity and drug consumption trends in the two groups. A strongly statistically significantly higher proportion of BMS patients (133; 55%) suffered from essential HTN compared to that in the controls (81, 33.5%) (*p*-value: <0.001 **). Specifically, 22 BMS patients were not aware that they were suffering from HTN, which was firstly detected through the above screening and subsequently assessed by the specialist. Instead, no differences were detected with respect to the other systemic diseases. With regard to drug intake, a higher prevalence of BMS patients were taking drugs compared with that of the controls (175, 72.3%; 139, 57.4%). A statistically significantly higher number of BMS patients was in treatment with antiplatelets (66; 27.3%) compared to that of the healthy subjects (34;14%) (*p*-value: <0.001 **). No difference was found in antihypertensive drug consumption between the patients and controls. 

Table 3 and Figure 2 show the frequency distribution of HTN in the BMS patients and controls, considering the age ranges and gender. There was a significant difference in the prevalence of HTN between the patients and controls at different ages; indeed, a higher prevalence of HTN was found in the BMS males older than 75 and in the male controls aged between 65 and 75 years, respectively. Instead, in the female patients and controls, a higher prevalence was found at the same age (65–75). It is worth noting that there was a higher prevalence of HTN in the BMS patients, both males and females, older than 75 (40% and 30.88% respectively) compared with that in the controls (19.05% and 17.95%, respectively).

The prevalence of oral symptoms and oral sites involved is shown in Table 4. By definition, all BMS patients reported a burning sensation (242; 100%). The other oral symptoms most frequently reported were xerostomia (149; 61.6%), followed by dysgeusia (110; 45.6%), globus pharingeus (78; 32.2%), intraoral foreign body sensation (53; 21.9%), sialorrhea (47; 19.4%) and a subjective change in tongue and gum morphology (42; 17.4%). Moreover, 102 (42.1%) BMS patients complained of generalized symptoms affecting all oral sites. In order, the tongue (211; 87.2%) was the most affected site, followed by the anterior palate (144; 59.5%), the lips (142; 58.7%), the gums (142; 58.7%) and the checks (139; 55.6%). 

The results of the dependence analysis between HTN and the qualitative and quantitative predictors in the BMS patients and controls are summarized in Table 5. A positive correlation was found between HTN and employment status (*p*-value: <0.001 **), systemic diseases (*p*-value: <0.001 **), drug consumption (*p*-value: 0.002 **) and level of education (*p*-value: 0.002 **) in the BMS patients. Instead, in the healthy controls, HTN was correlated only with systemic diseases and drug consumption (*p*-value: <0.001 **). Specifically, 130 (97.7%) BMS patients suffered from other systemic diseases, in addition to HTN, with only three patients affected solely by HTN.

The hierarchical multiple regression analyses predicting HTN in the BMS patients and controls are shown in Table 6. Regarding the BMS patients, in the first model (sociodemographic variables, risk factors and BMI), only age was found to be statistically significant and resulted in a significant increase in the R^2^ value (14.2%). The addition of the clinical parameters of pain (NRS, SF-MPQ), anxiety (HAM-A), depression (HAM-D) and sleep (PSQI) in the second model resulted in a significant increase in the R^2^ value (DR^2^ = 1.5%; *p*-value: <0.001 **). Moreover, the addition of systemic diseases (model 3) and drug consumption (model 4) resulted in a significant increase in the R^2^ values (DR^2^ = 8.9%, *p*-value: <0.001 **; DR^2^ = 2.3%, *p*-value: 0.006 ** respectively). The final full model (model 5), in which all variables were entered simultaneously, could explain 11.3% of the variance in HTN. Therefore, age, anxiety, systemic diseases and drug consumption were predictors of HTN in the BMS patients, but age and the presence of other comorbidities may be considered the most important in accordance with model 5.

With respect to the controls, in the first model (sociodemographic variables, risk factors and BMI), age, smoking and BMI were found to be statistically significant and resulted in a significant increase in the R^2^ value (7.2%). The addition of NRS, SF-MPQ, HAM-A, HAM-D, PSQI AND ESS (model 2) resulted in a significant increase in the R^2^ value (DR^2^ = 3.2%; *p*-value: 0.004 **). The addition of systemic diseases (model 3) and drug consumption (model 4) resulted in a significant increase in the R^2^ values (DR^2^ = 16.6%, *p*-value: <0.001 **; DR^2^ = 11.8%, *p*-value: 0.006 ** respectively). The final full model (model 5), in which all the variables were entered simultaneously, could explain 20.1% of the variance in HTN (*p*-value: <0.001 **). Therefore, age, smoking, BMI, systemic diseases, drug consumption and years of education were predictors of HTN with the presence of other comorbidities considered the most important.

## 4. Discussion

BMS is a chronic neuropathic orofacial pain condition in which the proportion of elderly patients with several comorbidities is consistently rising, making the assessment and treatment increasingly complex [15,16]. Anxiety, depression and sleep disturbances are the most frequent comorbidities associated with BMS [13,14,48,49]. However, recently, Canfora et al. have also found a high prevalence of cognitive decline and WMCs in BMS patients compared with that in a control group of healthy subjects [50].

Moreover, this high prevalence of WMCs in the brain of patients with BMS has been further confirmed in a more extensive study suggesting a potential role for cardiovascular risk factors, especially HTN, in the brain’s aging and subsequently in aggravating the disease [27].

Until now, the role of comorbidities in BMS has been poorly studied and understood, but in a recent study, de Pedro et al. reported that BMS patients suffer from a worsened health status, consuming more medications with a worse general quality of life, requiring an interdisciplinary therapy approach [8].

Therefore, the primary outcome of this study was to establish if there is any difference in terms of the prevalence of one of the most common medical comorbidities, such as HTN, between BMS patients and control subjects in order to confirm these previous studies findings.

The results of this study have indeed shown a statistically significant difference in the prevalence of HTN in BMS patients compared with that in healthy controls. HTN was found in 55% (133) of the BMS patients and in 33.5% (81) of the controls. Specifically, HTN was found in 65 males (48.8%) and in 68 females (51.3%) with BMS, and considering the age stratification, the highest prevalence of HTN was found in males older than 75 years of age (26; 40%) and in females between 65 and 75 years of age (27; 39.71%). 

In the different age groups, the prevalence of HTN varied between the cases and controls, mainly in males, with a higher prevalence of HTN in the controls between 65 and 75 years (21; 50%). A considerable difference in the prevalence of HTN was found in the elderly patients with BMS (>75) compared with that in the controls (male BMS patients: 26, 40% versus male controls: 8; 19.05%; female BMS patients: 21, 30.88% versus female controls: 7, 17.95%). In addition, taking into account the median age of the patients and controls (65 years), the prevalence of HTN is in line with epidemiological studies for controls (33.5%) but it is higher in BMS patients, reaching 55%. 

The prevalence of HTN in our sample of patients is slightly higher than the rate reported in a previous study by Bruehl et al. [24] on 300 chronic pain patients (39%) but is similar to the findings of a large Australian study on patients attending in pain clinics where the prevalence of HTN among chronic pain patients was 5.6 to 9.8 times higher compared with that in the general population [51]. 

This high prevalence of HTN in BMS patients may be explained considering that chronic pain and HTN often co-occur and share several biological and lifestyle risk factors, with functional and bidirectional interactions between the cardiovascular system and pain modulatory pathways. Indeed, blood pressure is modulated by functional circuitry linking the hypothalamus, nucleus tractus solitarius, nucleus raphe magnus and rostral ventrolateral medulla, involving the activity of the central adrenergic fibers and alpha-2 receptors, which in turn may modulate the descending activity of the anti-nociceptive pathways [52,53]. The prolonged activation of the descending adrenergic inhibitory pathways in patients suffering from BMS [54], especially in those with a long-lasting disease or in older patients, where aging naturally affects the intra-cortical connectivity of the brain resulting in an alteration to the modulatory system in the control of pain, may contribute to a blood pressure regulation system impairment, increasing the risk of HTN [55,56].

Moreover, it is notable that the pathogenesis of HTN is influenced by environmental and genetic factors, with these latter characteristics seemingly being responsible for 30% to 50% of any differences in blood pressure among individuals [6,57]. Therefore, it is possible to consider that the interaction between pain modulation and blood pressure may also explain why decreased metabolism of norepinephrine to normetanephrine, due to specific variants of catechol-O-methyltransferase (COMT) with an increase in the cerebral catecholamines, which can contribute to a higher blood pressure, may also modify the pain response [58,59,60,61]. Indeed, the gene that encodes this protein activity is highly polymorphic, and carriers of a variant, such as the homozygous met (A) allele, show a significant decrease in COMT activity, with a higher blood pressure, resulting in a higher prevalence of HTN compared with that in Met/Val or Val/Val carriers [59].

From the multivariate regression analysis, it is noteworthy that the predictors were slightly different with regard to the BMS patients and controls. Indeed, HTN was associated with advancing age, anxiety symptoms, systemic comorbidities and drug consumption in the BMS patients, while smoking, BMI, and years of education, in addition to advancing age, the occurrence of systemic comorbidities and drug consumption, were predictors of HTN in the controls. In the last models, where all variables were entered simultaneously, only age and the presence of systemic comorbidities remained as predictors of HTN in BMS patients, while educational level, smoking, the presence of systemic comorbidities and drug consumption were predictors in the controls, in line with the epidemiological studies [1,5,57]. It is surprising to consider that common predictors of HTN, such as obesity, smoking and a lower education level, were significant in relation to the controls but not in the BMS patients. This result may support the hypothesis that the onset of HTN in BMS may depend on other factors related to the effect of a painful disease on the brain [52,62].

In this study, age was a stronger predictor of HTN in the BMS patients than in the controls, which may explain the higher prevalence of HTN in older patients with BMS. Although, the effect of age was lost in the final regression model of the controls, it is known that advancing age is a common and universal predictor of a physiological increase in blood pressure, as suggested by epidemiological studies that reported a progressive increase in blood pressure of ≈7 mmHg per decade in subjects over 40 years of age, reaching a pressure of 140 mmHg during the eighth decade [63]. Nevertheless, the aging effect, as a predictor of HTN, may be stronger with conditions of chronic pain, such as BMS, which may already have a negative impact on the brain, above and beyond the fact that age-related effects cause accelerated brain aging [56,62]. Indeed, a previous study has suggested that chronic pain is associated with an “older” brain, at least two years older than the chronological age [64].

These results may confirm the findings of previous studies on the multifactorial origin of HTN, where brain alterations may have a role in initiating HTN, in addition to common risk factors, suggesting that the brain is either an early target of the disease or that the disease may also originate in the brain [4,65,66].

As a result, taken together, HTN and BMS may have an additional effect on cerebral health causing further morphological and chemical alterations in the brain and a reduction in the grey matter volume in specific brain areas, such as the prefrontal, medial frontal, inferior temporal and cerebellar areas, progressing as far as cerebral atrophy [67,68]. Brain aging, in turn, may further affect blood pressure regulation and worsen pain perception, aggravating both diseases [69].

It is notable that 16.5% (22) of the BMS patients were not aware that they were suffering from HTN. Therefore, it may be useful not only to consider the history of the patient but also to evaluate blood pressure to detect and treat unknown cases of HTN in order to reduce the long-term effect of HTN with respect to any further impairment of the brain in BMS patients.

In line with previous studies, the coexistence of other systemic comorbidities represents a strong predictor of HTN in patients and in controls [70,71]. Specifically, it is known that HTN rarely presents itself, but rather that it is a manifestation of a metabolic syndrome, characterized by a combination of elevated blood pressure, hypercholesterolemia, insulin resistance and abdominal obesity [5,72].

In this study, after covariate adjustment, psychological distress was not significantly associated with the development of HTN. However, it is important to consider that among mood disorders, anxiety was a predictor of HTN in a single model of the regression. This result is in line with previous studies, which have demonstrated a bidirectional association between anxiety and HTN. Indeed, individuals with HTN are more prone to have anxiety and those with anxiety are at risk of developing HTN, independently from other risk factors for HTN [73,74]. Moreover, it is known that in a long-term stressful lifestyle, events may act on the hypothalamic-pituitary-adrenal (HPA) axis, manifesting as anxiety or a somatic symptom, such as BMS [75]. The activation of the HPA axis, in turn increasing circulating catecholamines, may contribute to the development of a cardiovascular disease, such as HTN [75,76,77].

In contrast with the research of Giummara et al. [51], no correlation between HTN and pain was found in the BMS patients in our study, confirming the results of another study in which the association between pain and HTN may be mediated by shared risk factors, such as obesity and a lower educational level [78].

### Limitations

Firstly, an important limitation of the study is related to the hypertension diagnosis because it was not possible to verify if hypertension preceded the chronic pain onset or on the contrary, if the chronic pain came first. This could be important considering the etiopathogenesis of interpretation. In addition, being an explorative study, we matched the patients taking into account only age and sex, but in further studies, it will be necessary to match patients and controls also considering risk factors and BMI. Secondly, the assumption of the duration of antihypertensive drugs and eventually switching therapy could not be reported by the patients, and as a consequence, we do not know if there were patients suffering from resistant hypertension; consequently, although no correlation was found with HTN and pain scores, it is not possible to address the question of whether controlling the high pressure may have a role in modulating pain perception. Finally, the recruitment of the participants was undertaken in tertiary referral Oral Medicine Units, and this may not represent the real analysis of all patients affected by BMS, which are predominantly women (female/male ratio 3:1) [7].

## 5. Conclusions

In our research, the BMS patients showed a higher prevalence of HTN compared with that in the controls, and common predictors, such as smoking, higher BMI and a lower educational level, did not contribute to HTN in the BMS patients. The onset of HTN in BMS patients is not completely clear, but it may have a multifactorial etiopathogenesis in which genetic, biological and environmental factors could act in a synergic way. The co-occurrence and the long-term effects of HTN and painful disease on patients may in turn accelerate brain aging, aggravating both diseases and increasing the risk of premature mortality. Therefore, routine screening for the early detection and proper treatment of HTN may be considered appropriate in patients suffering from BMS. Although the role of dental professionals is essential in the diagnosis of BMS, it seems equally important to request an interdisciplinary care plan to ensure not only adequate pain management but also the integrated treatment of any comorbidities in order to improve the prognosis and quality of life of such patients. Further studies are required to confirm the association between HTN and BMS and to evaluate the impact of an integrated care plan, focused on the treatment of pain, on comorbidities and also lifestyle and health risk factors in BMS.

## Figures and Tables

**Figure 1 ijerph-20-02040-f001:**
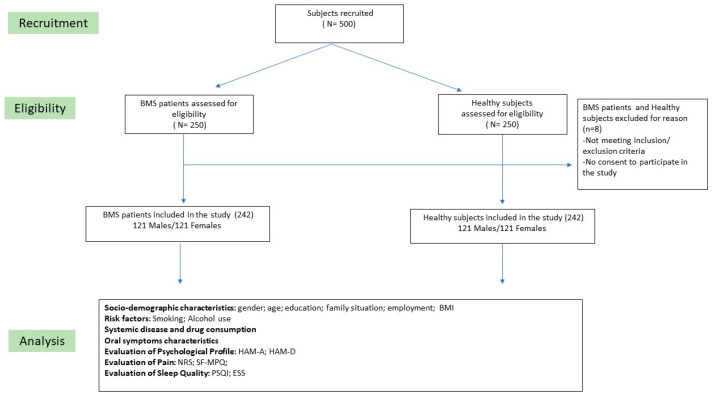
Flow Chart of the study; Abbreviations: BMS: burning mouth syndrome; BMI: body mass index; HAM-A: Hamilton rating scale for anxiety; HAM-D: Hamilton rating scale for depression; NRS: numeric rating scale; SF-MPQ: short-form Mc-Gill pain questionnaire; PSQI: Pittsburgh sleep quality index; ESS: Epworth sleepiness scale.

**Figure 2 ijerph-20-02040-f002:**
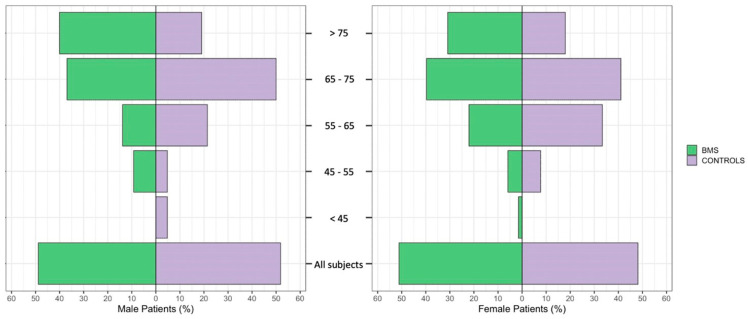
Frequency distribution of HTN by age ranges and gender among 133 BMS patients and 81 controls.

**Table 1 ijerph-20-02040-t001:** Socio-demographic profile, risk factors and clinical parameters of 242 BMS patients and 242 controls.

**Demographic variables**	**BMS**	**CONTROLS**	***p*-value**
**Gender**	**Frequency (%)**	**Frequency (%)**	1.000
**Male**	121 (50)	121 (50)	
**Female**	121 (50)	121 (50)	
**Age** (in years)	**Mean ± SD**	**Mean ± SD**	0.231
	65.61 ± 12.7	64.34 ± 10.4	
**Education** (in years)	**Mean ± SD**	**Mean ± SD**	**<0.001 ****
	9.31 ± 4.49	11.6 ± 4.92	
**Family situation**	**Frequency (%)**	**Frequency (%)**	0.254
Single	14 (5.8)	22 (9.1)	
Married	182 (75.2)	184 (76)	
Divorced	14 (5.8)	14 (5.8)	
Widowed	32 (13.2)	21 (8.7)	
**Employment**	**Frequency (%)**	**Frequency (%)**	**0.002 ****
Employed	68 (28.1)	101 (41.7)	
Unemployed	73 (30.2)	46 (19)	
Retired	101 (41.7)	95 (39.3)	
**Risk factors**	**Frequency (%)**	**Frequency (%)**	***p*-value**
**Smoking**			**0.007 ****
Never	181 (74.8)	176 (72.7)	
<5 cigarettes	8 (3.3)	21 (8.7)	
5–10 cigarettes	8 (3.3)	16 (6.6)	
10–15 cigarettes	17 (7)	16 (6.6)	
>15 cigarettes	28 (11.6)	13 (5.4)	
**Alcohol use**			**<0.001 ****
Never	191 (78.9)	149 (61.6)	
Yes (1 unit)	36 (14.9)	64 (26.4)	
Yes (2 units)	11 (4.5)	23 (9.5)	
Yes (>2)	4 (1.7)	6 (2.5)	
**Body Mass Index (kg/m^2^)**			**<0.001 ****
BMI < 18.5	0 (0)	6 (2.5)	
BMI: 18.5–24.9 normal	67 (27.7)	122 (50.4)	
BMI: 25.0–29.9 overweight	139 (57.4)	94 (38.8)	
BMI: 30–34 class I obesity	30 (12.4)	18 (7.4)	
BMI: 35–39.99 class II obesity	4 (1.7)	2 (0.8)	
BMI > 40 class III obesity	2 (0.8)	0 (0)	
**BMI**	**MEAN ± SD** 27.1 ± 3.48	**MEAN ± SD** 24.8 ± 3.48	
**Clinical parameters**	**BMS** **Median; IQR**	**CONTROLS** **Median; IQR**	***p*-value**
**NRS**	10 [10–10]	0 [0–0]	**<0.001 ****
**SF-MPQ**	11 [7–12]	0 [0–0]	**<0.001 ****
**HAM-D**	16 [14–20]	5 [2–10]	**<0.001 ****
**HAM-A**	17.5 [15–20]	5 [2–9]	**<0.001 ****
**PSQI**	8 [8–10]	5 [3–8]	**<0.001 ****
**ESS**	7 [5–9]	5 [3–8]	**<0.001 ****

The significance difference between means was measured by performing a *t*-student test. ** Significant *p* ≤ 0.01. The significance of differences between percentages was measured by performing the Pearson Chi Square test. ** Significant *p* ≤ 0.01. IQR is the interquartile range. The significance of differences between medians was measured by performing the Mann–Whitney test. ** Significant with Bonferroni correction 0.008. Abbreviations: BMI: body mass index, BMS: burning mouth syndrome, ESS: Epworth Sleepiness Scale; HAM-A: Hamilton rating scale for anxiety; HAM-D: Hamilton rating scale for depression, NRS: numeric rating scale; PSQI: Pittsburgh sleep quality index; SF-MPQ: short-form McGill pain questionnaire.

**Table 2 ijerph-20-02040-t002:** Evaluation of the prevalence of systemic diseases, drug consumption and antihypertensive drugs in 242 BMS patients and 242 controls.

**Systemic diseases**	**BMS** **Frequency (%)**	**CONTROLS** **Frequency (%)**	***p*-value**
Hypertension	133 (55)	81 (33.5)	**<0.001 ****
Hypercholesterolemia	83 (34.3)	69 (28.5)	0.203
Gastrointestinal diseases	43 (17.8)	32 (13.2)	0.209
Hypothyroidism	29 (12)	22 (9.1)	0.375
Prostatic hypertrophy	23 (9.5)	11 (4.5)	0.049
Other cardiovascular diseases	23 (9.5)	18 (7.4)	0.514
Myocardial infarction	16 (6.6)	8 (3.3)	0.141
Neoplastic diseases	13 (5.4)	19 (7.9)	0.361
Respiratory diseases	10 (4.1)	10 (4.1)	1.000
HCV infection	3 (1.2)	7 (2.9)	0.339
HBV infection	2 (0.8)	1 (0.4)	1.000
Neurological disorders	2 (0.8)	6 (2.5)	0.285
Hyperthyroidism	1 (0.4)	3 (1.2)	0.623
Endocrine diseases	1 (0.4)	3 (1.2)	0.623
**Drug consumption**	**BMS** **Frequency (%)**	**CONTROLS** **Frequency (%)**	***p*-value**
Antiplatelets	66 (27.3)	34 (14)	**<0.001 ****
Proton pump inhibitors	53 (21.9)	39 (16.1)	0.132
Statins	50 (20.7)	51 (21.1)	1.000
Beta blockers	42 (17.4)	36 (14.9)	0.537
ACE-inhibitors	38 (15.7)	35 (14.5)	0.800
Angiotensin II receptor antagonists (ARBs)	37 (15.3)	18 (7.4)	0.009
Thiazide diuretics	27 (11.2)	27 (11.2)	1.000
Calcium channel blockers	24 (9.9)	13 (5.4)	0.086
Levothyroxin sodium	24 (9.9)	17 (7)	0.327
Blood thinners	16 (6.6)	5 (2.1)	0.023
Bisphosphonates	5 (2.1)	4 (1.7)	1.000
Steroids	3 (1.2)	5 (2.1)	0.724

A significant difference between the percentages was measured by performing Fisher’s exact test. ** Significant *p* ≤ 0.01. Significant with Bonferroni correction 0.003 for the systemic diseases and for the drug consumption.

**Table 3 ijerph-20-02040-t003:** Frequency distribution of HTN by age range and gender of 133 BMS patients and 81 controls.

Male Patients			Female Patients	
BMS	CONTROLS		BMS	CONTROLS
Frequency (%)	Frequency (%)	Age	Frequency (%)	Frequency (%)
65 (48.87)	42 (51.85)	All subjects	68 (51.13)	39 (48.15)
0 (0)	2 (4.76)	<45	1 (1.47)	0 (0)
6 (9.23)	2 (4.76)	45–55	4 (5.88)	3 (7.69)
9 (13.85)	9 (21.43)	55–65	15 (22.06)	13 (33.33)
24 (36.92)	21 (50)	65–75	27 (39.71)	16 (41.03)
26 (40)	8 (19.05)	>75	21 (30.88)	7 (17.95)

**Table 4 ijerph-20-02040-t004:** Prevalence of oral symptoms and sites involved in 242 BMS patients.

**Oral symptoms**	**BMS** **Frequency (%)**
Burning	242 (100)
Xerostomia	149 (61.6)
Dysgeusia	110 (45.6)
Globus pharingeus	78 (32.2)
Intraoral foreign body sensation	53 (21.9)
Sialorrhea	47 (19.4)
Subjective change in tongue and gum morphology	42 (17.4)
Itching	27 (11.2)
Tingling sensation	25 (10.3)
Oral dyskinesia	18 (7.4)
Occlusal dysesthesia	16 (6.6)
Halitophobia	14 (5.8)
Dysosmia	5 (2.1)
**Sites involved**	**BMS** **Frequency (%)**
**Generalized**	102 (42.1)
Tongue	211 (87.2)
Anterior palate	144 (59.5)
Lips	142 (58.7)
Gums	142 (58.7)
Cheeks	119 (49.4)
Soft palate	110 (45.5)
Floor of the mouth	107 (44.2)

**Table 5 ijerph-20-02040-t005:** Dependence analysis of 242 BMS patients and 242 controls with HTN and qualitative and quantitative Predictors.

**Qualitative variables**	**BMS/HTN** **Frequency (%)**		***p*-value**	**Controls/HTN** **Frequency (%)**		***p*-value**
**Marital status** **Married** **not married**	**98 (73.3)** **35 (26.3)**		0.554	64 (79) 17 (21)		0.524
**Employment** Employed Not employed	25 (18.8) 108 (81.2)		**<0.001 ****	26 (32.1) 55 (67.9)		0.038
**Smoking** Smoker Non-smoker	15 (11.3) 118 (88.7)		0.347	29 (35.8) 52 (64.2)		0.185
**Alcohol use** Yes No	29 (21.8) 104 (78.2)		0.874	23 (28.4) 58 (71.6)		0.384
**Systemic diseases** Yes No	130 (97.7) 3 (2.3)		**<0.001 ****	77 (95.1) 4 (4.9)		**<0.001 ****
**Drug consumption** Yes No	107 (80.5) 26 (19.5)		**0.002 ****	68 (84) 13 (16)		**<0.001 ****
**Quantitative variables**	**BMS/HTN** **Median; IQR**	**BMS/NO HTN** **Median; IQR**	***p*-value**	**Control/HTN** **Median; IQR**	**Control/NO HTN** **Median; IQR**	***p*-value**
**NRS**	10 [10–10]	10 [9–10]	0.545	0 [0–0]	0 [0–0]	0.401
**SF-MPQ**	11 [7–12]	11 [7–12]	0.792	0 [0–1]	0 [0–0]	0.408
**HAM-D**	16 [14–20]	16 [13–20]	0.799	5 [3–11]	4 [2–9]	0.049
**HAM-A**	18 [15–21]	17 [15–20]	0.383	6 [3–12]	4 [2–9]	0.034
**PSQI**	8 [8–10]	8 [8–10]	0.416	6 [3–10]	4 [3–7]	0.017
**ESS**	7 [6–9]	7 [5–9]	0.642	6 [4–9]	5 [2–8]	0.074
**Education** **(in years)**	8 [5–13]	9 [8–13]	**<0.001 ****	13 [8–16]	13 [8–14]	0.936
**BMI** **(kg/m^2^)**	27.3 [25.6–28.8]	26.1 [24.7–28.3]	0.026	25.6 [23–27.6]	23.9 [21.9–26.3]	0.010

A significant difference between the percentages was measured by performing Fisher’s exact test. ** Significant with Bonferroni correction 0.003 for qualitative predictors. IQR is the interquartile range. The significance of differences between medians was measured by performing the Mann–Whitney test. ** Significant with Bonferroni correction 0.006 for quantitative predictors. *Abbreviations*: BMI: body mass index; BMS: burning mouth syndrome, ESS: Epworth sleepiness scale; HAM-A: Hamilton rating scale for anxiety; HAM-D: Hamilton rating scale for depression; HTN: hypertension, NRS: numeric rating scale; PSQI: Pittsburgh sleep quality index; SF-MPQ: short-form McGill pain questionnaire.

**Table 6 ijerph-20-02040-t006:** Multivariate logistic regression analysis predicting HTN in 242 BMS patients and 242 controls.

**Predictors of HTN** **in BMS patients**	**Model 1**		**Model 2**		**Model 3**		**Model 4**		**Model 5**	
	** *OR* **	** *p-value* **	** *OR* **	** *p-value* **	*OR*	*p-value*	*OR*	*p-value*	*OR*	*p-value*
**Age**	1.08	**<0.001 ****	1.08	**<0.001 ****	1.07	**<0.001 ****	1.08	**<0.001 ****	1.07	**<0.001 ****
**Gender: Male**	0.90	0.740	0.86	0.653	0.84	0.606	1.04	0.899	0.88	0.721
**Years of education**	0.95	0.165	0.95	0.193	0.95	0.135	0.96	0.290	0.95	0.168
**Marital status: Married**	0.83	0.601	0.79	0.513	0.67	0.296	0.75	0.429	0.61	0.217
**Job: Employed**	1.14	0.740	1.12	0.783	1.11	0.800	1.09	0.826	1.07	0.871
**Smoker**	0.86	0.748	0.85	0.732	0.98	0.967	0.75	0.555	0.85	0.771
**Alcohol use**	0.83	0.616	0.83	0.629	0.74	0.439	0.80	0.563	0.73	0.431
**BMI**	1.06	0.147	1.06	0.177	1.07	0.151	1.06	0.176	1.06	0.228
**NRS**			1.04	0.767					1.05	0.711
**SF-MPQ**			0.99	0.837					0.99	0.780
**HAM-D**			0.92	0.099					0.91	0.094
**HAM-A**			1.10	**0.050 ***					1.08	0.118
**PSQI**			0.94	0.334					0.91	0.200
**ESS**			1.04	0.497					1.07	0.260
**Systemic diseases**					16.26	**<0.001 ****			14.76	**<0.001 ****
**Drug Consumption**							2.51	**0.007 ****	1.89	0.101
** *R^2^ (%)* **		**<0.001 ****		**<0.001 ****		**<0.001 ****		**<0.001 ****		**<0.001 ****
** *R^2^ change (%)* **				0.524		**<0.001 ****		**0.006 ****		**<0.001 ****
**Predictors HTN in control subjects**	**Model 1**		**Model 2**		**Model 3**		**Model 4**		**Model 5**	
	** *OR* **	** *p-value* **	** *OR* **	** *p-value* **	*OR*	*p-value*	*OR*	*p-value*	*OR*	*p-value*
**Age**	1.04	**0.034 ***	1.04	**0.040 ***	1.04	0.056	1.04	**0.049 ***	1.04	0.066
**Gender: Male**	1.22	0.516	1.34	0.360	1.39	0.346	1.48	0.250	1.63	0.183
**Years of education**	1.03	0.315	1.03	0.336	1.09	**0.024 ***	1.07	0.075	1.09	**0.028 ***
**Marital status: Married**	1.34	0.409	1.49	0.286	1.49	0.304	1.28	0.509	1.44	0.361
**Job: Employed**	0.63	0.223	0.55	0.132	0.87	0.764	0.72	0.448	0.72	0.491
**Smoker**	2.63	**0.007 ****	2.41	**0.017 ***	2.38	**0.029 ***	2.79	**0.009 ****	2.64	**0.022 ***
**Alcohol use**	0.52	0.071	0.62	0.203	0.51	0.106	0.49	0.070	0.58	0.216
**BMI**	1.09	**0.042 ***	1.09	0.069	1.11	**0.029 ***	1.09	0.077	1.09	0.101
**NRS**			0.99	0.960					0.94	0.653
**SF-MPQ**			1.12	0.356					1.06	0.639
**HAM-D**			0.98	0.656					0.98	0.636
**HAM-A**			1.03	0.535					0.99	0.826
**PSQI**			1.05	0.280					1.04	0.434
**ESS**			1.06	0.183					1.08	0.113
**Systemic diseases**					22.17	**<0.001 ****			11.72	**<0.001 ****
**Drug Consumption**							7.56	**<0.001 ****	2.97	**0.014 ***
** *R^2^ (%)* **		**0.005 ****		**0.004 ****		**<0.001 ****		**<0.001 ****		**<0.001 ****
** *R^2^ change (%)* **				0.124		**<0.001 ****		**<0.001 ****		**<0.001 ****

SE are the standard errors of the beta estimates. The *p*-values were obtained from the hypothesis test on the regression coefficients. * Moderately significant 0.01 < *p*-value ≤ 0.05. ** Strongly significant *p*-value ≤ 0.01. *Abbreviations*: BMI: body mass index; BMS: burning mouth syndrome, NRS: numeric pain intensity scale; T-PRI: total pain rating index; HAM-A: Hamilton rating scale for anxiety; HAM-D: Hamilton rating scale for depression; HTN: hypertension, PSQI: Pittsburgh quality index; ESS: Epworth sleepiness scale.

## Data Availability

Data are available by contacting the corresponding author.

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
