# Peer review of "Burning Mouth Syndrome and Hypertension: Prevalence, Gender Differences and Association with Pain and Psycho-Social Characteristics—A Case Control Study"

_ijerph, 2023, doi:10.3390/ijerph20032040_

Round 1

Reviewer 1 Report

The study does not specify how the sample size was calculated.

It is not stated how the matching between cases and controls was performed.

the last references do not exist or are incorrectly placed

It is not clear which are the categories of references to carry out the interpretation of the OR

Author Response

  • The study does not specify how the sample size was calculated.

Thank you for your comment. The sample size, equal to 242 patients for each group (the BMS patients and controls), was calculated to obtain a power test value (1- Beta) at no less than 99%, associated with a significance of no more than 1%. This sample size was obtained using the effect size value equal to 0.65, measured in a previously published research study regarding the Age Related White Matter Changes and correlation with BMS. [27]  [lines 188-192]

It is not stated how the matching between cases and controls was performed.

Thank you for your comment. We Based on the average age and sex distribution of the BMS patients, control subjects were then enrolled using a convenient sampling in order to obtain controls matched by sex and age. Then, the sample size was calculated as described in Statistical Methods section.

  • The last references do not exist or are incorrectly placed

Thank you for your comment. We have delayed these typing errors.

  • It is not clear which are the categories of references to carry out the interpretation of the OR

Thank you for your comment. We have added the following reference “Bland JM, Altman DG. Statistics notes. The odds ratio. BMJ. 2000 May 27;320(7247):1468. doi: 10.1136/bmj.320.7247.1468. PMID: 10827061; PMCID: PMC1127651. [lines 206-209]

Reviewer 2 Report

Authors preformed case control study to assess prevalence of BMS in HT patients.  The study was written well however. Authors condsens the paragraphs with little spaces among the lines making following it very difficult.  Method should be categorized into subheadings such as study design,  inclusion criteria, data collection  etc. Authors should clarify how they calculated sample size of this study. Conclusion was reasonable 

Author Response

  • Authors preformed case control study to assess prevalence of BMS in HT patients.  The study was written well however. Authors condsens the paragraphs with little spaces among the lines making following it very difficult. 

Thank you for your comment. We have increased the line spacing.

  • Method should be categorized into subheadings such as study design, inclusion criteria, data collection etc.

Thank you for your comment. We have added 6 subheadings for Methods.

  • Authors should clarify how they calculated sample size of this study. Conclusion was reasonable 

The sample size, equal to 242 patients for each group (the BMS patients and controls), was calculated to obtain a power test value (1- Beta) at no less than 99%, associated with a significance of no more than 1%. This sample size was obtained using the effect size value equal to 0.65, measured in a previously published research study regarding the Age Related White Matter Changes and correlation with BMS. [27]  [lines 188-192]

Reviewer 3 Report

A few corrections are required

(The Authors must see my remarks)

Author Response

Thank you for your comments. We have made your corrections. 

Reviewer 4 Report

Title:

Burning Mouth Syndrome and Hypertension: prevalence, gender differences and correlation with Pain and Psycho-social characteristics 

A very good study, few points of interest.

The title of the article is appropriately selected and denotes the study performed.

The abstract is concise enough meanwhile the aim is clear and the methodology explained briefly what was done.

The references are recent and relevant in addition to this are well written and arranged and highly related to the study. My comment about the references number 73 and 74 need to be rechecked again.

The authors reviewed well what is already written and investigated in similar studies and in addition to this the research question was clearly outlined and properly justified and related to what is already known about the topic.

The weakest point in the study what the authors mentioned in the limitation: who came first the hypertension or the chronic pain, this was possible to be addressed during the case selection procedures and the cases enrolment?!

The methodology is clear enough and the number of participating subjects is fair for both groups.

The results are well written and the tables are relevant although there is some text repetition, better to reduce as much as possible. 

The discussion section is very well written and elaborated good with other researchers results.

Author Response

My comment about the references number 73 and 74 need to be rechecked again.

Thank you for your comments. We have delayed these typing errors.

 The weakest point in the study what the authors mentioned in the limitation: who came first the hypertension or the chronic pain, this was possible to be addressed during the case selection procedures and the cases enrolment?!

Thank you for your comments. We considered this point a part of the limitations of our study because we could not know exactly the patients’ hypertension onset, but we could only know when the antihypertensive treatment started. Even these we do not have any data about the correlation in terms of time between the hypertension onset and the pain onset. We would like to study this process in future studies considering the hypertension also as a predictive factor for brain aging.

The methodology is clear enough and the number of participating subjects is fair for both groups. The results are well written and the tables are relevant although there is some text repetition, better to reduce as much as possible.

Thank you for your comments. We have modified the text in order to avoid any useless repetition.

Round 2

Reviewer 1 Report

The study is well organized congratulation!